# Non-Contact Radiofrequency Inductive Sensor for the Dielectric Characterization of Burn Depth in Organic Tissues

**DOI:** 10.3390/s19051220

**Published:** 2019-03-11

**Authors:** Thi Hong Nhung Dinh, Stéphane Serfaty, Pierre-Yves Joubert

**Affiliations:** 1Centre for Nanoscience and Nanotechnology (C2N), CNRS, University Paris-Sud, Université Paris-Saclay, C2N-Orsay, 91405 Orsay, CEDEX, France; thi-hong-nhung.dinh@u-psud.fr; 2Systems and Applications of Information and Energy Technologies (SATIE), CNRS, Université de Cergy-Pontoise, 95000 Cergy-Pontoise, France; stephane.serfaty@u-cergy.fr

**Keywords:** dielectric characterization, burn wounds, organic tissue, non-contact sensors, radiofrequency sensors, inductive sensors

## Abstract

A flat circular transmission line-based 300 MHz resonator was implemented for the non-contact assessment of burn depths in biological tissues. Used as a transmit-and-receive sensor, it was placed at a 2 mm distance from organic material test samples (pork fillet samples) which were previously burned on their surface in various heating conditions involving different temperatures, durations, and procedures. Data extracted from the sensor by means of a distant monitoring coil were found to clearly correlate with the depth of burn observed in the tissue samples (up to 40% sensor output changes for a 7 mm burn depth) and with the heating conditions (around 5% sensor output changes observed in samples burned with identical heating procedures but at two different temperatures—75 °C and 150 °C—and around 40% sensor output changes observed between samples heated at the same temperature but with different heating procedures). These results open the way for the development of easy-to-implement assessment and monitoring techniques for burns, e.g., integrated in wearable medical dressing-like monitoring devices.

## 1. Introduction

The timely and accurate assessment of burn wound depth has raised great interest in medicine, since it has a significant impact on patient management outcome. Indeed, burn wound depth assessment is mostly carried out by clinical evaluation relying on visual and tactile characterizations [1]. However, it is commonly accepted that the depth of burn wounds is not entirely assessable immediately after the injury. First, the extent of tissue damage may not be immediately visible [2], and secondly, according to a physiopathological process known as burn wound conversion [3], a superficial partial thickness burn may progress into deeper tissue in the initial few days following the injury. This process is of clinical consequence in the treatments of burn wounds. Indeed, unclear burn depth assessments may lead to a delay in relevant treatments such as surgical debridement and skin grafting of deep dermal and full-thickness burns, resulting in increased hypertrophic scarring, additional risks of complications such as cellulitis, late return of the patient to daily life, and late recovery of pre-injury functions [1]. Therefore, there are benefits from developing sensing methods that are able to quantitatively and continuously assess burn wound depths, aiding in carrying out a timely clinical decision-making. 

Burn depth quantitative assessments require the use of dielectric contact probes [4] or advanced imaging techniques [1] which possibly suffer from practical limitations when continuous bedside monitoring of burn wounds is required. In this paper, we investigate the relevance of a non-contact electromagnetic technique for the continuous assessment of burn wounds. The investigated method could lead to the development of low-cost and easy-to-implement instrumented medical dressings dedicated to the non-contact assessment and continuous monitoring of burn wounds. 

It is known that cutaneous thermal burns and wounds result in the destruction of tissue cells and organic content, which induces noticeable changes in the local dielectric properties (DP) of the tissue [4,5,6]. The DP constitute the response of the tissue to an external electrical field. They are represented by the complex permittivity *ε**:(1)$$\varepsilon^{*}=\varepsilon \varepsilon_{0}+\frac{\sigma }{j\omega }$$
where *ε* is the relative dielectric constant of the material, *σ* is its electrical conductivity, *ω* = 2*πf* is the angular frequency, *f* is the frequency, *j* is equal to $\sqrt{-1}$ and *ε*_0_ is the dielectric constant of vacuum. The relative dielectric constant of the material *ε* measures the extent to which the electrical bound charge distributions within the tissue can be distorted and polarized under the influence of the electrical field, and the consequent energy stored within the tissue. These charge distributions are mainly associated with double-layer membrane proteins, the surface of solvated macromolecules, or other polar molecules [7]. The conductivity *σ* of the tissue, expressed in S/m, is relative to the dielectric losses associated with the displacement and conduction currents induced within the tissue. This is primarily due to the ionic free charges moving through the tissue under the influence of the field. All electrically active entities within the tissue exhibit their own characteristic response to the applied field, according to their own nature, shape, and size. As a result, the macroscopic DPs (*ε* and *σ*) exhibit frequency-dependent behaviors which have been observed and studied in a 1 Hz–100 GHz frequency bandwidth [8,9,10,11]. Since burn wounds result in the alteration of tissue cells and in the loss of the extra- and intracellular liquids, both the dielectric constant and the conductivity are expected to change in the event of burns. One difficulty for developing devices able to sense these alterations lies in the choice of relevant investigation frequencies within the 1 Hz–100 GHz bandwidth which enable the changes of both the conductivity and the dielectric constant to be sensed with equivalent sensitivity.

Previous works have shown that radiofrequency (RF) electromagnetic inductive methods using wireless resonant RF antennas (i.e., operating at a single frequency) are good candidates for the non-contact sensing of DP changes in organic material, provided the frequency of investigation is properly chosen [12]. Indeed, the proportion of the dielectric losses to the stored energy is given by the loss tangent, so that: (2)$$\tan\delta =\frac{\sigma }{\varepsilon_{0}\varepsilon_{}\omega }$$

In order to sense structural modifications of biological material through changes of their macroscopic dielectric properties, it is relevant to choose the frequency of investigation at which the changes of the real part and the imaginary part of *ε** are both significant, i.e., of the same order of magnitude. This condition is reached for tan*δ* = 1. As a result, for a given tissue, the ideal sensing angular frequency *ω* would be:(3)$$\omega =\sigma /\varepsilon_{0}\varepsilon_{}$$

In this paper, an inductive resonant RF probe is implemented and evaluated for the non-contact sensing of thermally induced burns in organic test samples made out from low-fat pork fillet pieces. The paper is organized as follows. In Section 2, the principle of operations of the used probe is presented together with the probe data extraction method. In Section 3, the preparation of the burned tissue samples, the experimental set up, and the implementation of the used probe are presented. Then, in Section 4, the obtained sensing results are presented and discussed. Section 5 presents the conclusions.

## 2. Principle of the Contactless RF Sensing Technique

The probe used in this study is constituted of (i) a flat RF high-Q resonator acting as a non-contact transmit-and-receive sensor which radiates the electromagnetic field within the tissue, and (ii) a distant monitoring coil connected to a vector network analyzer (VNA), constituting a wireless sensor data “reader” able to sense the resonator impedance changes (Figure 1a). Knowing that the used tissue samples are muscle tissue, they are featured by *ε* = 60 and *σ* = 1 S/m [8] around 300 MHz. As a result, Equation (3) leads to the choice of an investigation frequency close to 300 MHz, i.e., so that tan*δ* = 1 for that kind of tissue. Please note that for burn depth assessment on actual skin tissue, the choice of the resonant frequency be different, since the dielectric parameters for skin tissue are of different values (e.g., *σ_skin_* ≈ 0.3 S/m and *ε_skin_* ≈ 100 in the 50–500 MHz bandwidth [13]) leading to a different investigation frequency according to Equation (3) (*f_skin_* ≈ 54 MHz for the given values of *σ_skin_* and *ε_skin_*). Here, considering the nature of the tissue samples used in this study, the used RF resonator is designed to feature a resonant frequency of 300 MHz. 

The used resonator is a flat 19 mm-diameter multi-turn split conductor transmission-line resonator (MTLR) similar to RF antennas used in the context of magnetic resonance measurement techniques [14]. It is constituted of two rolled-up 1 mm-width transmission lines constituted of 35 µm-thick copper tracks deposited by photolithography on each side of a 250 µm-thick low-loss dielectric substrate (CuFlon). Due to its geometrical and material features, the used MTLR exhibits a measured load-free resonance frequency *f*_0_ = 302.6 MHz and a measured quality factor Q ≈ 300. It is modelled by means of an equivalent electric model featuring resistive, inductive, and capacitive lumped elements denoted *R*_1_, *L*_1_, and *C*_1_, respectively, with *R*_1_ = 710 mΩ, *L*_1_ = 109 nH, and *C*_1_ = 2.55 pF [15]. When placed at a distance *d*_2_ from the tissue sample, additional electrical lumped elements, namely, resistance *R_i_* and inductance *L_i_*, are added to the single-mesh electrical model of the MTLR. These additional elements take into account the complex dielectric properties of the tissue sample. More precisely, *R_i_* represents the electromagnetic energy transmitted by the MTLR which is dissipated within the tissue. It is related to the electrical conductivity of the tissue. *L_i_* represents the energy that is stored within the tissue, which is related to its dielectric constant. 

The dependence of *R_i_* on the electrical conductivity and the dependence of *L_i_* on the dielectric constant were theoretically established in reference [12] for volume RF resonators such as a multi-gap parallel plate cylindrical resonator. More precisely, *R_i_* was found to be proportional to the tissue conductivity, and *L_i_* was found to be proportional to the dielectric constant of the tissue. These conclusions were experimentally confirmed in reference [16]. These features were also computationally assessed for flat circular antennas such as MTLR in reference [17] and experimentally verified with reference liquids used as test samples in another study [18]. Here, the monitoring coil is a 8 mm-diameter, 11 mm-height copper–wire bobbin coil, modelled by resistance *R_c_* (50 Ω) in series with inductance *L_c_* (0.22 µH). The bobbin coil is placed at a distance *d*_1_ = 16 mm from the MTLR. It is electromagnetically coupled to the MTLR (Figure 1a) and connected to the VNA. The MTLR is placed at a distance *d*_2_ = 2 mm from the sample. Finally, the coupling between the monitoring coil, the MTLR, and the tissue under evaluation is modelled by the equivalent circuit depicted in Figure 1b. 

Considering the electrical model of the probe presented in Figure 1b, the impedance *Z_mes_* measured at the end by the monitoring coil of the loaded RF probe reads:(4)$$Z_{mes}=Z_{c}+\frac{C_{1}k^{2}L_{c}\left(L_{1}+L_{i}\right)j\omega^{3}}{1+C_{1}\left(R_{1}+R_{i}\right)j\omega +C_{1}\left(L_{1}+L_{i}\right){\left(j\omega \right)}^{2}}$$

In practice, a minimization algorithm such as the damped Gauss–Newton iterative minimization algorithm [19] can be used to fit the model of Equation (4) with the experimental data of *Z_mes_* acquired in a frequency bandwidth *B* centered on the resonance frequency of the sensor. In the case of the unloaded resonator (*R_i_* and *L_i_* being equal to 0), the fitting allows *R*_1_, *L*_1_, and *C*_1_ to be estimated, knowing *Z_c_* and the resonance frequency *f*_0_ [15]. Using the same estimation technique, it is possible to read out the impedance of the resonator interacting with the tissue, thus estimating *R_i_* and *L_i_* after prior estimation of *R*_1_, *L*_1_, and *C*_1_. Here, *R_i_* and *L_i_* were estimated using data measured in a 2 MHz bandwidth around the observed resonance frequency of the loaded resonator.

## 3. Materials and Methods

### 3.1. Tissue Samples

In this study, a series of tissue samples were prepared so as to evaluate the feasibility of burn depth sensing using the MTLR probe. The tissue samples were cut out of a low-fat piece of pork fillet, to form a set of samples of similar shapes. Small samples featured approximatively a 5 cm side length, a 2 cm height, and a mass of 30 g. Large samples featured approximatively a 10 cm side length, a 2 cm height, and a mass of 60 g. 

### 3.2. Tissue Heating Procedures

The samples were burned on one face by contact with a plate heated at 75 °C or 150 °C, with heating durations *t_H_* = 30, 60, 120, and 240 s. Also, two different heating procedures were used. In the continuous heating procedure (CHP), the samples were heated from *t* = 0 s to *t* = *t_H_* without interruption (Figure 2). The CHP samples were then cooled down to ambient temperature (25 °C) and measured using the RF probe. In the interrupted heating procedure (IHP), the samples were heated first from *t* = 0 s to *t* = 30 s, cooled down, and measured, and then heated again for 30 s, cooled down, and measured, and so on (Figure 3). Figure 4 shows examples of samples prepared with IHP.

### 3.3. Experimental Set Up and Measurement Procedure

Figure 4a shows the RF measurement set up which was composed of (i) the RF probe (RF resonator and monitoring coil) and the sample to be tested, installed on a mechanical support, (ii) a thermo-regulated chamber in which the probe and the sample are placed during measurement, and (iii) a HP4195A network analyzer. The analyzer was equipped with a 41951A impedance test kit, which allows the RF probe impedance to be directly read out and recorded by the analyzer in a 100 kHz–500 MHz bandwidth.

Here, a 20 MHz bandwidth BW centered on the expected RF probe resonance frequency, i.e., BW = 295–315 MHz, was considered for the measurements. In this bandwidth, the port of the network analyzer was first calibrated with short, open, and matched 50 Ω loads in sequence. Then, a 0 Ω compensation procedure was carried out in order to directly sense the equivalent impedance of the loaded RF resonator. The procedure consisted in measuring the impedance of the monitoring probe alone (*Z_c_*) in the 292–312 MHz bandwidth and subtracting it from the impedance of the loaded RF probe (*Z_mes_*) in order to directly record the equivalent impedance of the resonator coupled with the dielectric sample, *Zmes − Zc* (see Equation (4)).

The RF probe was implemented at 25 °C. The measured equivalent impedance of the resonator (*Z_mes_ − Z_c_*) is presented in Figure 5a, with the resonator loaded with a raw sample, a 240 s-IHP burned sample, or left unloaded. One can note the sensitivity of the RF probe to the various samples. Moreover, Figure 5b shows the measured impedance *Z_mes_ − Z_c_* versus frequency in modulus and phase when the RF resonator was loaded with a burned sample, which featured a resonance frequency close to 305.5 MHz. 

## 4. Results and Discussion

### 4.1. Samples Heated with Interrupted Heating Procedures (IHP)

The relative variations of *R_i_* and *L_i_*, estimated from data measured in IHP samples heated at a 150 °C temperature, are shown in Figure 6. Each point on these graphs results from the averaging of nine estimations carried out on three different samples prepared with the same procedure; the error bars are the standard deviation of the estimations. Furthermore, each type of sample was cut after the RF measurements in order to estimate the actual burn depth by means of microscope observations. *R_i_* and *L_i_* changes in the burned tissues were significant, with differences of −30% and −40%, respectively, observed in a sample heated at 150 °C for 240 s. 

The observed decrease of *R_i_* with the heating time is consistent with the loss of extra- and intracellular liquids induced by the heating of the tissue. Indeed, this loss of liquid reduces the ionic conduction and, therefore, reduces the value of *R_i_* which is proportional to the tissue conductivity (see Section 2). Besides, the observed decrease of *L_i_* with the heating time is consistent with the destruction of cell membranes. Indeed, cell membrane destruction results in the decrease of the dielectric constant which is proportional to *L_i_*. Also, these changes clearly correlated with the depth of the burns observed in the tissue samples, which reached 7 mm for a 240 s heating duration.

In order to study the influence of the heating temperature on the sample burns, small samples (30 g) were heated during a unique 60 s heating process (CHP) at various temperatures selected in the 75 °C to 250 °C range. The observed dielectric changes are presented in Figure 7. One can note that for a given heating duration, the heating temperature clearly influenced the tissue alterations observed through the changes of *R_i_* an *L_i_*, especially in the 75 °C to 175 °C temperature range. In order to further study the influence of the temperature on the tissue burns, similar experiments were conducted on large samples (60 g) heated with an IHP at 75 °C and 150 °C. One can note in Figure 8 that results similar to the results presented in Figure 6 were obtained: large decrease of *R_i_* and *L_i_* with the time of heating and strong correlation to the burn depth. However, one can also notice that the different heating temperatures actually induced different dielectric change patterns in the *L_i_* -versus-*R_i_* representation space of Figure 8b. This seems to indicate that the loss of tissue water (observable in the *R_i_* changes) and the destruction of the cell membranes (mostly observable in the *L_i_* changes) are two temperature-dependent tissue structural changes, the proportions of which being also dependent on the temperature of the heating process. In these experiments, it can be noted that even if 240 s heating processes implemented at 75 °C and 150 °C led to identical burn depths of 5.5 mm in the 60 g samples, the *R_i_*-changes-versus-*L_i_*-changes curves of Figure 8b show that the probe was sensitive to the temperature-dependent alteration processes of the tissue. 

### 4.2. Continuous Heating Procedure (CHP) versus IHP

CHP and IHP were carried out at 150 °C and up to 240 s in small (30 g) samples. The resulting burned samples were measured using the proposed non-contact sensing method. The obtained results are presented in Figure 9. It can be observed that the used heating procedures resulted in different burn depth after a 240 s heating duration (9 mm for CHP and 7 mm for IHP). Furthermore, CHP burns featured large *R_i_* changes (over 50% reduction) and smaller *L_i_* changes (over 30% reduction). 

This behavior was different in the case of IHP (*R_i_* reduction was 30%, and *L_i_* reduction was 40%). From these observations, it can be presumed that the tissue alterations processes (loss of tissue liquid and cell membrane destruction) are concurrent phenomena which are dependent on the heating procedure in addition to being dependent on the chosen temperature. Furthermore, the used IHP and CHP heating procedures led to different burn depths, in addition to different *R_i_* and *L_i_* changes. As a result, burn depth assessments and possibly deeper tissue characterizations could be envisaged from *R_i_* and *L_i_* measurements by means of suited signal processing methods such as machine learning algorithms. 

## 5. Conclusions

The implementation of a non-contact RF probe was carried out for the sensing of burns made on low-fat pork fillet samples, used as organic tissue test samples. The proposed method appears to be relevant to sense the complex dielectric changes of burned tissues, through the changes of complex impedance of the RF probe induced by the changes of the dielectric properties of the burned samples. Also, the influence of the tissue heating conditions on the tissue burns, which involved different temperatures and/or procedures, was studied. The obtained results showed that the sensing of both the conductivity and the dielectric constant of the tissue are relevant to characterize tissue burns. Associated with dedicated inverse problem-solving algorithms (e.g., machine learning algorithms) the proposed method could be considered to actually assess the depth of burns in tissues induced by various heating conditions. Also, tailor-made low-cost resonator readers can be considered to be used instead of laboratory VNA for low-cost implementation purposes. These preliminary results open the way for promising non-contact diagnostics and the monitoring of burn wounds using MTLR sensors, which, owing to their planar geometry and wireless implementation, could be integrated in smart medical dressings to constitute wearable and disposable burn wound monitoring devices.

## Figures and Tables

**Figure 1 sensors-19-01220-f001:**
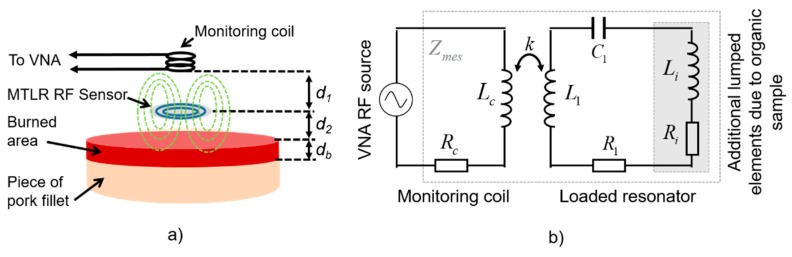
(**a**) Schematic representation of the set up; (**b**) Modeling of the setup using an equivalent electrical circuit.

**Figure 2 sensors-19-01220-f002:**
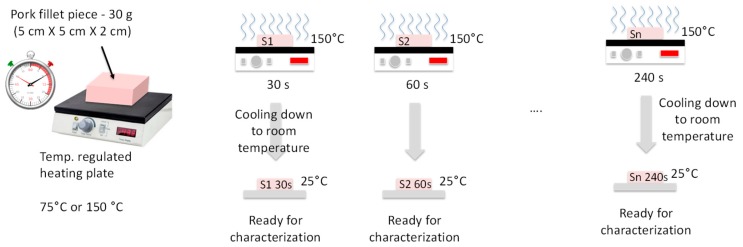
Continuous sample-heating procedure (CHP).

**Figure 3 sensors-19-01220-f003:**
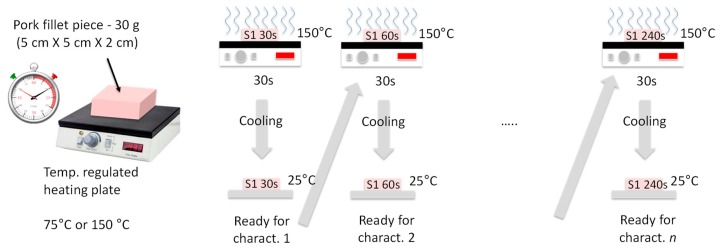
Interrupted sample-heating procedure (IHP).

**Figure 4 sensors-19-01220-f004:**
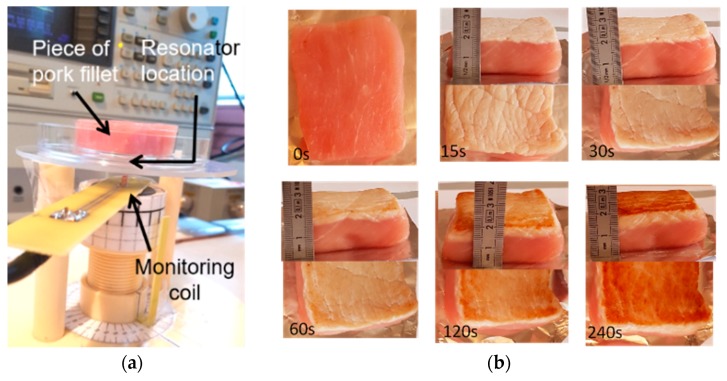
(**a**) Radiofrequency (RF) measurement set up; (**b**) Examples of low-fat pork filet samples, raw and burned during up to 240 s using the interrupted heating procedure.

**Figure 5 sensors-19-01220-f005:**
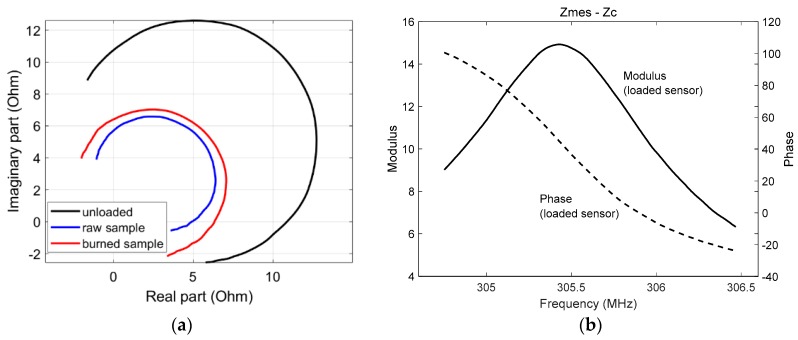
(**a**) Equivalent impedance *Z_mes_* − *Z_c_* obtained when the RF probe was unloaded and loaded with the raw and burned samples. The impedance changes are presented in the complex plane in a 304.6–306.4 MHz frequency bandwidth; (**b**) Modulus and phase of the equivalent impedance *Z_mes_ − Z_c_* of the RF resonator when loaded with the sample versus frequency in a 304.6–306.4 MHz bandwidth.

**Figure 6 sensors-19-01220-f006:**
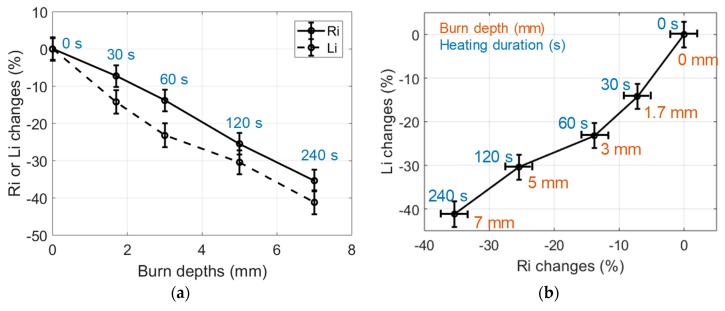
(**a**) Changes of the dielectric properties of 150 °C IHP-samples expressed in % vs burn depths. Heating durations were 0, 30, 60, 120, 240 s; (**b**) Changes of the dielectric properties of 150 °C IHP-burned samples expressed in % in the (*R_i_*, *L_i_*) impedance plane (30 g samples).

**Figure 7 sensors-19-01220-f007:**
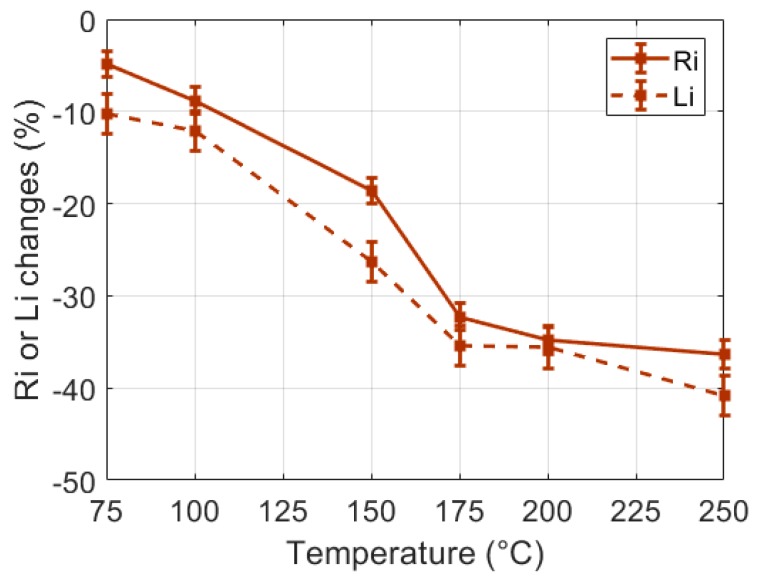
Changes of the dielectric properties of 60 s CHP (30 g) samples versus the heating temperature.

**Figure 8 sensors-19-01220-f008:**
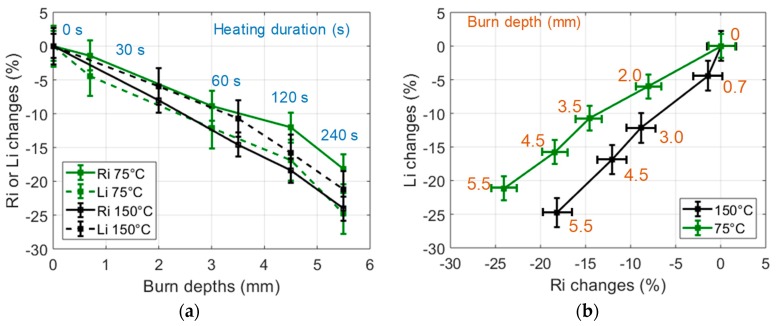
(**a**) Changes of the dielectric properties of 75 °C and 150 °C IHP-samples expressed in % vs burn depths. Heating durations were 0, 30, 60, 120, 240 s; (**b**) Changes of the dielectric properties of 75 °C and 150 °C IHP-samples expressed in % in the (*R_i_*, *L_i_*) impedance plane (60 g samples).

**Figure 9 sensors-19-01220-f009:**
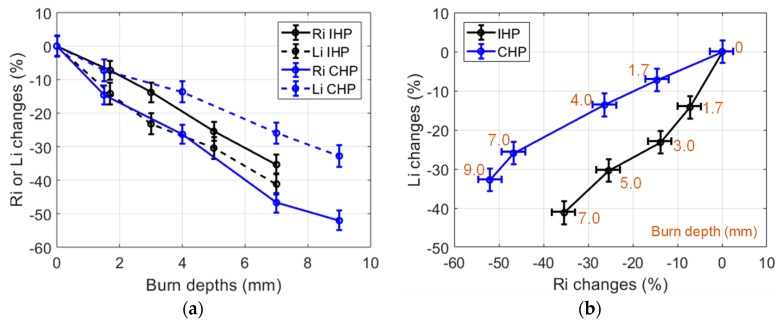
(**a**) Changes of the dielectric properties of 150 °C IHP-samples and CHP-samples expressed in % vs burn depths. Heating durations were 0, 30, 60, 120, 240 s; (**b**) Changes of the dielectric properties of 150 °C IHP-samples and CHP-samples expressed in % in the (*R_i_*, *L_i_*) impedance plane (30 g samples).

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
