# Peer review of "Non-Contact Radiofrequency Inductive Sensor for the Dielectric Characterization of Burn Depth in Organic Tissues"

_sensors, 2019, doi:10.3390/s19051220_

Round 1

Reviewer 1 Report

The work is interesting. Its presents a non-invasive method is presented to detect the depth of burns.

I miss a brief description of the different sections at the end of the introduction.

Please see the details in the attachment.

Author Response

The authors thank the reviewer for his time and efforts to help them improving the paper.

The response to the reviewer's comment are provided in the attached file.    

Reviewer 2 Report

This paper presents a non-contact RF inductive sensor for burn depth assessment. The approach is interesting and the paper overall fits well with the scope of the MDPI Sensors. Additionally, the paper is well organized, the methods are presented with appropriate justification. Results for burn depth in pork fillet samples are promising. However, to make conclusive comments, measurements need to be expanded. Below are further specific comments:

-          Line 29. The motivation needs to be elaborated further. What are some of these significant impacts on patient management outcome?

-          Line 127: Are dielectric parameters of the samples characterized? If not, this information is already mentioned in Line 78. If these values are based on measurements, methods for how the impedance measurements were conducted should be detailed.

-          The results are on low-fat muscle tissue. Considering the application, it is expected for the authors comment on changes in the design parameters of the RF sensor explained in Section 2 with the inclusion of other tissues in a realistic burn scenario such as fat tissue and skin tissue.

-          Line 179: Burnt tissue results in loss of fluids and thus ionic conduction, therefore it is expected to see an increase in the Resistance. Why are plots in Figures 6, 7, and 8 demonstrate the opposite?

-          Line 191: This comment is way too early to make based on measurements in only two temperatures. Therefore, I ask the authors to make more temperature measurements. Also, when presenting the results in Fig. 7, the authors should comment on why two different slopes are expected theoretically for different temperature burns.

-          Line 204: Another early comment. My comment above applies for the results on two different heating procedures. More measurements are needed. Additionally, the proper justification of the results should be made when presenting it in Fig. 8.

Author Response

The authors warmly thank the reviewer for his time and efforts to help them to improve the paper.  

Please find the reponse to the reviewer in the attached file.

Round 2

Reviewer 2 Report

Authors addressed my comments.